# Real-World Experience of Measurable Residual Disease Response and Prognosis in Acute Myeloid Leukemia Treated with Venetoclax and Azacitidine

**DOI:** 10.3390/cancers14153576

**Published:** 2022-07-22

**Authors:** Shin Yeu Ong, Melinda Tan Si Yun, Nurul Aidah Abdul Halim, Dheepa Christopher, Wei Ying Jen, Christian Gallardo, Angeline Tan Hwee Yim, Yeow Kheong Woon, Heng Joo Ng, Melissa Ooi, Gee Chuan Wong

**Affiliations:** 1Department of Haematology, Singapore General Hospital, Singapore 169856, Singapore; melinda.tan.s.y@singhealth.com.sg (M.T.S.Y.); nurul.aidah.abdul.halim@singhealth.com.sg (N.A.A.H.); angeline.tan.h.y@singhealth.com.sg (A.T.H.Y.); woon.yeow.kheong@sgh.com.sg (Y.K.W.); ng.heng.joo@singhealth.com.sg (H.J.N.); wong.gee.chuan@singhealth.com.sg (G.C.W.); 2Department of Haematology, Tan Tock Seng Hospital, Singapore 169856, Singapore; dheepa_christopher@ttsh.com.sg (D.C.); christian_aledia_gallardo@ttsh.com.sg (C.G.); mdcogmm@nus.edu.sg (M.O.); 3Department of Haematology-Oncology, National University Cancer Institute, Singapore General Hospital, Singapore 169856, Singapore; wei_ying_jen@nuhs.edu.sg

**Keywords:** acute myeloid leukemia, venetoclax, measurable residual disease

## Abstract

**Simple Summary:**

More than 90% of patients with acute myeloid leukemia carry aberrant markers that can be detected using flow cytometry. Monitoring low levels of residual disease after treatment (measurable residual disease) is an important biomarker to assess the efficacy of treatment and can identify patients who may eventually relapse. Our retrospective study aimed to determine the potential value of MRD for prognosticating outcomes in AML patients treated with less intensive chemotherapy. In 63 patients with newly diagnosed AML, we found that detectable residual disease by flow cytometry was associated with a higher incidence of relapse, and shorter progression-free and overall survival. We also confirmed that the threshold of measurable disease affecting outcomes is 0.1%. MRD is useful in patients to monitor AML patients treated with less intensive therapy.

**Abstract:**

The prognostic value of measurable residual disease (MRD) by flow cytometry in acute myeloid leukemia (AML) patients treated with non-intensive therapy is relatively unexplored. The clinical value of MRD threshold below 0.1% is also unknown after non-intensive therapy. In this study, MRD to a sensitivity of 0.01% was analyzed in sixty-three patients in remission after azacitidine/venetoclax treatment. Multivariable cox regression analysis identified prognostic factors associated with cumulative incidence of relapse (CIR), progression-free survival (PFS) and overall survival (OS). Patients who achieved MRD < 0.1% had a lower relapse rate than those who were MRD ≥ 0.1% at 18 months (13% versus 57%, *p* = 0.006). Patients who achieved an MRD-negative CR had longer median PFS and OS (not reached and 26.5 months) than those who were MRD-positive (12.6 and 10.3 months, respectively). MRD < 0.1% was an independent predictor for CIR, PFS, and OS, after adjusting for European Leukemia Net (ELN) risk, complex karyotype, and transplant (HR 5.92, 95% CI 1.34–26.09, *p* = 0.019 for PFS; HR 2.60, 95% CI 1.02–6.63, *p* = 0.046 for OS). Only an MRD threshold of 0.1%, and not 0.01%, was predictive for OS. Our results validate the recommended ELN MRD cut-off of 0.1% to discriminate between patients with improved CIR, PFS, and OS after azacitidine/venetoclax therapy.

## 1. Introduction

Measurable residual disease (MRD) is an established independent indicator of disease recurrence after intensive chemotherapy in acute myeloid leukemia (AML) and may identify patients who benefit from treatment intensification [1,2,3]. In a meta-analysis of 81 studies reporting on 11,151 adults with AML treated after intensive induction chemotherapy, patients with MRD negativity had improved rates of overall survival (OS) (68% versus 34%) and relapse-free survival (RFS) (64% versus 25%) at 5 years compared to patients with MRD positivity [4]. In older AML patients who are ineligible for intensive induction chemotherapy, venetoclax and hypomethylating agents (HMA) have been established as a new standard of care, leading to high rates of remission with MRD negative responses (23.4% versus 7.6% with just azacitidine) [5]. However, data regarding the role of MRD monitoring in patients with AML treated with lower-intensity treatment regimens remain limited.

To date, some clinical trial data suggest that MRD measured by multiparameter flow cytometry (MFC) retains its prognostic value when applied to older adults treated with lower intensity frontline regimens of HMA and venetoclax. A phase II trial of 10-day decitabine with venetoclax showed that MRD negativity at 2 and 4 months was associated with longer RFS and OS [6]. MRD data were also recently reported from the phase III trial of azacitidine and venetoclax, showing that patients with undetectable MRD had a longer remission duration, event-free survival (EFS) and OS [7]. To our knowledge, no study has analyzed the prognostic impact of “real-world” MFC MRD assessment after azacitidine and venetoclax treatment in newly diagnosed AML patients. Previous studies have shown that patients treated with the same regimen in real-world settings have worse outcomes than in clinical trials, possibly due to differences in comorbidities, protocol adherence, and centre experience or expertise [8,9]. Furthermore, a retrospective analysis of older patients treated with azacitidine showed that MRD negativity was not associated with improved OS due to the competing risk of death and infectious complications in a frail patient population [10]. In addition, MRD assessment in an off-trial setting may suffer methodological heterogeneity with reduced sensitivity and specificity, thus reducing its prognostic significance [11].

Another relevant but relatively unexplored area of investigation is the predictive MRD threshold for patients undergoing semi-intensive chemotherapy. While the ELN Working Party advocates the 0.1% threshold level to distinguish between MRD positivity and negativity [12], MRD levels < 0.1% may still indicate active disease and a threshold of 0.035% has been prospectively validated in AML patients receiving intensive induction therapy [13]. The clinically relevant MRD threshold is likely impacted by the intensity of chemotherapy delivered as well as characteristics of the MRD test, especially when performed by flow cytometry, where varying techniques and expertise can affect the quality and interpretation of flow data [14,15]. Further research is needed with regard to the clinical implication of MFC MRD levels below 0.1% after semi-intensive therapy.

Herein, we investigated the role of MFC MRD in refining complete response in elderly AML patients treated with azacitidine and venetoclax in a real-world setting from three tertiary institutions in Singapore. A secondary aim is to investigate the clinically relevant threshold for MRD after semi-intensive therapy.

## 2. Methods

### 2.1. Patients and Treatment

The Singhealth Duke-NUS, Tan Tock Seng Hospital, and National University Hospital Institutional Review Boards approved the study to analyze patient data retrospectively. Data from patients with newly diagnosed non-acute promyelocytic AML who received azacitidine and venetoclax off trial between 1 January 2019 and 1 August 2020 were collected.

### 2.2. Treatment, Monitoring and Follow Up

Azacitidine 75 mg/m^2^ was administered subcutaneously on days 1 to 7 of each 28-day cycle. Intravenous hydration and an oral uric acid lowering agent constituted tumour lysis syndrome prophylaxis. If antifungal prophylaxis with posaconazole was used, venetoclax dose was reduced by at least 75% [16], with no need to ramp-up dosing. If antifungal prophylaxis was not used, dose ramp-up to 200 mg (with ciprofloxacin or other moderate CYP3A inhibitors) or 400 mg (without ciprofloxacin) was performed with inpatient monitoring. Subsequent cycles of venetoclax and azacitidine were repeated as appropriate, and azacitidine dose modifications were used if myelosuppression persisted despite the dose reduction of venetoclax [17]. Discontinuation of azacitidine, and dose modification or discontinuation of venetoclax occurred at the discretion of the treating physician.

### 2.3. Response Assessments

Bone marrow biopsies were obtained at baseline, at the time of assessment of morphological response, and periodically after with MRD testing by flow cytometry. Response assessments used 2017 ELN criteria [18], and patients who achieved complete remission (CR), complete remission (CRi) with incomplete recovery of normal neutrophil or platelet counts, or morphologic leukemia-free state (MLFS) were included in the study if they had evaluable MFC MRD data. CR was defined as 5% blasts or less within the bone marrow and adequate peripheral blood counts (neutrophils  ≥  1.0 × 10^9^/L, platelets  ≥  100 × 10^9^/L).

MRD analysis by MFC was performed in a standardized manner following EuroFlow Standard Operating Procedures (SOPs) for flow cytometer settings, fluorescence compensation, sample preparation, and immunophenotyping protocols at respective laboratories of the three institutions [19]. The marrow aspirates were analyzed by at least a five tube, eight-color EuroFlow AML panels, consisting of CD45, CD34, CD117, and HLA-DR as the backbone markers in all tubes, and CD123, CD36, CD71, CD64, CD35, CD11b, C14, CD7, CD56, CD38, CD33, CD13, CD19, CD15, and CD4 across the remaining tubes [20]. Residual disease was identified using a strict gating strategy by experienced clinicians across the three institutions, according to the ELN leukemia-associated immunophenotype (LAIP) and different from normal (DfN) approach. A total of 1–5 × 10^5^ cells were acquired per tube after excluding debris. Detection threshold was to a sensitivity of 0.01%. MRD negative specimens with suboptimal cell counts were excluded. While respective laboratories follow EuroFlow SOPs, flow cytometry data were not shared between institutions, and interlaboratory validation was not performed to determine concordance of MRD results between each laboratory.

As none of the patients positive for MRD at response converted to MRD-negative state on repeat evaluations beyond the 3-month post-remission time point, the 3-month post-CR/CRi was chosen as the second time-point for MRD assessment. Patients who were tested for MRD more than once were required to maintain a sustained MRD negative state to be deemed MRD negative at the 3-month timepoint.

### 2.4. Statistical Analysis

Progression-free survival (PFS) was defined as time from start date of chemotherapy to date of disease progression, or death from disease or other causes, and overall survival (OS) was defined as time from start date of chemotherapy to date of death from any cause or date of last follow-up in surviving patients. Probabilities of time to relapse or death were estimated using the Kaplan–Meier method, and survival curves were compared between groups using the log-rank test. The cumulative incidence of relapse (CIR) was evaluated with the use of the method of Gray, and the Fine and Gray model for competing risks was measured from the date of MRD assessment until the date of relapse. Patients not known to have relapsed were censored on last examined date and patients who died without relapse were counted as a competing cause of failure. Multivariable time-dependent Cox regression models were constructed for CIR, PFS, and OS. Transplant after azacitidine/venetoclax therapy was incorporated as a categorical time-dependent covariate switching at the time of transplant. We used a stepwise approach to select confounding factors into the multivariable model, including variables with *p*-value < 0.2 in the univariable analysis.

## 3. Results

### 3.1. Patient Characteristics

Between 1 January 2019 and 1 August 2020, we treated 119 patients with newly diagnosed AML with azacitidine and venetoclax off trial. A total of 63 (53%) patients achieved a response and were included in this analysis. The median age was 65 years old (range 37 to 87), 51% had ELN adverse-risk disease, and 27% had complex cytogenetics, defined as karyotypes with ≥ 3 chromosomal abnormalities [21]. Of the 14 patients with FLT3-ITD/TKD mutation, 1 patient received a FLT3 inhibitor (midostaurin), in addition to azacitidine and venetoclax. Of 63 patients, 15 (24%) patients achieved CR, 42 (66.5%) achieved CRi, and 6 (9.5%) achieved MLFS as best response to azacitidine/venetoclax treatment. The median time to response achievement was 56 days (IQR 34–71 days). A total of 24 (38%) patients achieved MRD < 0.1% at the time of response. The rate of MRD < 0.1% and corresponding response by treatment cycle is depicted in Figure 1A. Notably, the rates of MRD < 0.1% among patients achieving CR, CRi, and MFLS were not significantly different (40%, 36%, and 50%, respectively; *p* = 0.785), suggesting a lack of association between hematologic recovery and MRD response. The rate of MRD < 0.1% and MRD < 0.01% responses increased from remission to cumulative 3-month timepoint post-remission, suggesting that response can deepen while continuing azacitidine/venetoclax treatment (Figure 1B). A total of 20 patients underwent stem cell transplant after achieving a response, and 9 were MRD negative at 0.1% sensitivity before the transplant. The median OS of these responding patients was 9.6 months (IQR 4.7 to 16.5 months) and median PFS was 8.5 months (IQR 4.4 to 13.6 months). After a median follow-up of 14.6 months (IQR 9.7 to 21.5 months), 19 (30%) patients relapsed and 26 (41%) died. Patients received a median of 5 cycles of azacitidine and venetoclax.

### 3.2. Prognostic Value of MRD by MFC in Patients Achieving Response

Azacitidine and venetoclax offered high rates of negative MRD < 0.1% at response in patients with favorable ELN risk (72.7%) compared to intermediate/adverse ELN risk (30.8%, *p* = 0.009). A trend towards higher rates of MRD negativity in patients with *NPM1* mutation (63.6% versus 32.7%, *p* = 0.055) compared to *NPM1^WT^* patients was also observed. Prior exposure to HMA was associated with lower rates of MRD negative (0.1%) response (14.3% versus 44.9%, *p =* 0.038) (Table 1). Competing risk analysis for cumulative incidence of relapse with death as a competing event showed a lower relapse rate in patients achieving MRD < 0.1% at response compared to those who were MRD positive at 18 months (13% versus 57%, *p* = 0.006). Patients achieving CR/CRi who were MRD negative at the 0.1% threshold had a significantly longer PFS (median not reached (NR) versus 12.6 months, *p* = 0.042) and OS (median 26.5 months versus 10.3 months, *p* = 0.020). In contrast, using a lower MRD threshold of 0.01% at response, while retaining prognostic significance for PFS, could not define patients with improved OS outcomes (*p* = 0.182).

We further investigated the impact on survival outcomes according to MRD ≥ 0.1% versus MRD ≥ 0.01% to < 0.1% versus MRD < 0.01% at the cumulative 3-month timepoint post-remission. Compared to patients with MRD < 0.01% (*n* = 32), patients with MRD ≥ 0.01% to < 0.1% (*n* = 15) had poorer PFS and OS, but this did not reach significance (HR 1.79, *p* = 0.85 and HR 1.31, *p* = 0.74, respectively). Conversely, patients with MRD ≥ 0.1% at response had significantly worse PFS and OS (HR 16.26, *p* < 0.001 and HR 5.17, *p* = 0.034, respectively), compared to patients with an MRD burden of < 0.01%. Due to small numbers of patients who had MRD ≥ 0.01% to < 0.1% at response (*n* = 4), we were not able to perform similar analyses at response.

### 3.3. MRD Negativity at 0.1% Level Is an Independent Predictor for CIR, RFS, and OS

Univariable analyses of risk factors for progression and death were carried out, including age (≥65 years versus <65 years), blood count parameters (hemoglobin level and neutrophil and platelet count), bone marrow blasts percentage (<30%, 30–50%, and >50%), de novo versus secondary AML, gene mutations (*FLT3-ITD/TKD*, *NPM1*, and biallelic *CEBPA*), number of venetoclax cycles (<2 versus ≥2), complex karyotype, ELN risk stratification (adverse versus intermediate/favorable), and prior HMA treatment. Confounding factors with a *p*-value of < 0.2 were included in the multivariable model, which included complex karyotype (HR 3.29, *p* = 0.012 and HR 2.98, *p* = 0.007 for PFS and OS, respectively), ELN adverse risk group (HR 2.11, *p* = 0.137 and HR 1.75, *p* = 0.171 for PFS and OS, respectively), and transplant (HR 0.32, *p* = 0.071 and HR 0.30, *p* = 0.133 for PFS and OS, respectively). Multivariable analysis adjusting for ELN risk group, complex karyotype, and transplant showed that MRD ≥ 0.01% at response and 3 months post-remission were independently associated with increased CIR and worse PFS, but not with OS (Table 2). A comparison of outcomes using the MRD threshold of 0.1% yielded stronger correlations with CIR and PFS, as well as a statistically significant association with OS. Patients with detectable MRD ≥ 0.1% versus MRD < 0.1% at response had significantly lower 18-month OS rates at 35.1% versus 69.7%, *p* = 0.020 (Figure 2).

Landmark analysis was performed at 4.1 months after achieving CR, CRi, or MLFS, which was the median time from starting treatment to undergoing transplant. Five patients were excluded due to relapse (*n* = 1), death (*n* = 3), or lost to follow-up (*n* = 1). Patients who underwent a stem cell transplant had a longer median OS compared with patients who did not or could not undergo a transplant. The median OS after this landmark date was 4.2 months versus 20.1 months: HR 0.29, 95% CI 0.10–0.81, and *p* = 0.018.

### 3.4. MRD Negativity at Treatment Termination Predicts for Lower Risk of Relapse

Twelve patients stopped treatment due to adverse events and not due to death, progression, or transplant in our cohort. Among these patients, five patients were MRD negative at 0.1% sensitivity at a cumulative 3-month timepoint post-remission, and none relapsed at a median follow-up of 7.3 months post-treatment discontinuation (IQR 6.1 to 13.4 months). In contrast, seven patients were MRD > 0.1% upon stopping treatment, and all relapsed at a median of 4.2 months after stopping therapy (IQR 2.6 to 12.0 months) (*p* = 0.001).

## 4. Discussion

Although MRD is commonly used for risk stratification in AML practice, data supporting its use have mainly been in patients on intensive chemotherapy and generated in clinical trials. The current study provides the first “real-world” evidence on the predictive value of MRD assessment measured by flow cytometry on relapse and survival in AML patients receiving semi-intensive induction therapy with azacitidine and venetoclax. We confirm that patients in CR/CRi with undetectable MRD at the 0.1% threshold either at response or 3 months post-remission had significantly improved CIR, PFS, and OS compared to patients with detectable MRD. MRD retained its prognostic value in multivariable models including transplant and genetic factors. Our findings are in accordance with preliminary data from clinical trials that MRD positivity after non-intensive induction is also associated with poor outcomes with shorter PFS and OS [6,7,22].

In our study, 38.1% of patients achieved an MRD-negative response (0.1% sensitivity), which is almost equivalent to results from the confirmatory phase III VIALE-A trial reporting that 40.9% of patients achieved undetectable MRD response after azacitidine/venetoclax treatment [7]. With high rates of complete remission achievable after venetoclax-based regimens, there is renewed interest in defining the role of MRD assessment for prognosis in AML patients treated with less intensive therapy. Patients in the VIALE-A trial who achieved MRD-negative responses had longer duration of response, EFS, and OS, compared to patients who had MRD ≥ 0.1% at remission. Similarly, Maiti et al. demonstrated that patients who underwent venetoclax plus decitabine and had MRD-negative responses at 2 and 4 months had longer RFS, EFS, and OS compared to MRD-positive patients [6]. Finally, MRD by flow cytometry was also prognostic in patients who received azacitidine or low-dose cytarabine with fludarabine in the phase III PETHEMA/FLUGAZA clinical trial; patients with negative MRD had prolonged RFS and OS compared to patients with detectable MRD [22].

In line with previous trial data [6,7], we show that achieving MRD-negative response at 0.1% sensitivity independently predicted CIR, PFS, and OS after treatment with azacitidine and venetoclax in real-world patients. Patients who achieved MRD-negative responses (0.1% sensitivity) had improved survival with a prolonged median OS of 26.5 months. Improved survival is demonstrated even among patients with ELN intermediate and poor-risk AML, with an 18-month estimated OS of 70% versus 46% in patients who were MRD positive (HR 0.29, 95% CI 010–0.87, *p* = 0.027). Interestingly, among patients who terminated treatment electively in our study, patients who were MRD negative (0.1% sensitivity) at cumulative 3 months post-remission remained relapse-free at a median follow-up of 7 months post-therapy discontinuation. Similarly, a recent retrospective study of 29 patients who electively ceased venetoclax and HMA therapy reported that MRD negative status predicted for prolonged treatment-free remission duration ≥ 3 years [23]. While cautious interpretation is warranted due to the small sample size, these observations may support therapy de-escalation if the MRD-negative response is sustained. Such an approach is pertinent for continuous azacitidine/venetoclax regimens in view of frequent grade 3 and 4 infectious complications and associated myelotoxicity [24]. Confirmation of these findings in the context of a prospective randomized discontinuation trial will help to define the optimal duration of azacitidine/venetoclax treatment and inform treatment-free surveillance in patients with MRD-negative response.

In our study, an MRD threshold of 0.1% reached higher statistical significance to discriminate patients with different RFS, compared with an MRD threshold of 0.01%. In addition, OS was only associated with a higher MRD threshold of 0.1%. At a 0.01% MRD cut-off level, nine patients with detectable MRD remained in remission (“false positive”), while only five patients had false-positive MRD results at a higher cut-off of 0.1%. Conversely, the number of MRD negative patients who relapsed (“false negative”) were similar at both thresholds (*n* = 2). Potentially, less intensive therapy may be associated with a milder debulking effect leading to a higher MRD threshold that is clinically significant [14]. We postulate that these results may also be explained by the higher prevalence of underlying myelodysplasia in our cohort of older patients treated with azacitidine and venetoclax. Since some of the aberrant antigen expression that is defined as LAIP may also be observed in myelodysplasia, low level expression may be related to lower risk dysplastic clones rather than residual AML blasts [25]. Another possible explanation is that patients regarded as MRD positive using a 0.01% threshold were continued on more intensive courses of azacitidine/venetoclax treatment, which may lead to prolonged myelosuppression and more infectious complications and death in a frail population. Our results substantiate the ELN recommended cut-off of 0.1% to discriminate patients with significantly different CIR, PFS, and OS, but this remains to be validated in larger prospective series.

While flow cytometry is highly applicable in more than 90% of AML cases [26], there is limited harmonization and standardization which may limit sensitivity and specificity in real-world settings [11]. Assessment of MRD at our centers utilized an eight-color Euroflow platform that incorporates the markers recommended in the recent ELN consensus document for MFC MRD assessment in AML [27,28]. Clinicians graded MRD using both LAIP and DfN approaches. Using adequate fluorochromes, multiple aberrant markers, and a combined LAIP/DfN approach for MRD detection facilitated accurate reporting, despite heterogeneity of gating strategy and interpretation among reporting physicians. Further investigation of background levels of aberrant immunophenotypic expression in cell populations with underlying myelodysplasia is needed to increase MFC MRD specificity and sensitivity in older adults undergoing treatment for AML.

We observed that *NPM1*-mutated AML patients had favorable responses to azacitidine/venetoclax, with a trend towards higher rates of MRD negative response at 0.1% sensitivity (64% versus 33%) and a 12-month OS of 72% versus 57% in patients without *NPM1* mutated AML. Previous retrospective studies comparing induction regimens for *NPM1* mutated AML also showed that venetoclax with HMA was highly effective, achieving high rates of remissions with MRD negativity (75% versus 27% of patients receiving just HMA) and durable remissions [29,30]. Hence, for older patients with *NPM1*-mutated AML, venetoclax-based regimens are highly effective and could possibly be considered as a mutation-targeted treatment option [30].

Limitations of the current study include its retrospective nature and the small sample size which restricted statistical power, as well as relatively short follow-up. Consideration of MRD cutoffs below the 0.1% threshold may have been limited by power in this small cohort size. Secondly, cases were only graded by one clinician who may not be blinded to patient outcomes, which could lead to bias. While respective laboratories at the three institutions used harmonized sample preparation, flow cytometer settings, and antibody panels according to EuroFlow SOPs, manual gating is inevitably expert-dependent, and interpretation of flow cytometry may vary. Data should be shared between institutions in future to ensure concordance in MRD status and levels between sites. Thirdly, not all patients had follow-up marrow assessments after 3 months post-remission, and the possibility of late MRD-negative responses occurring well after clinical remission warrants further evaluation. Lastly, we did not perform molecular profiling of patients beyond the *FLT-3*, *NPM1*, and *CEBPA* mutations, and were not able to correlate our MFC MRD results with molecular biomarkers.

## 5. Conclusions

In conclusion, the combination of venetoclax and azacitidine is now an established standard of care in AML patients who are not eligible for intensive therapy, and our findings support the role of MRD assessment as an important predictor of outcomes after lower intensity therapy in the real-world setting. We further validated the predictive MRD cut-off level of 0.1%. Future studies tailoring treatment according to MRD results could potentially guide clinical decisions in elderly AM, moving away from a “one size fits all” to precision medicine approach.

## Figures and Tables

**Figure 1 cancers-14-03576-f001:**
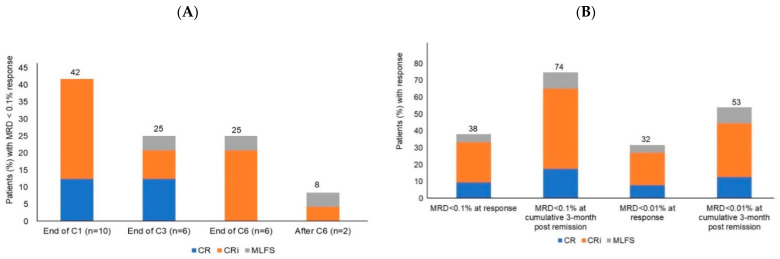
(**A**) MRD < 0.1% response by treatment cycle. End of C1 refers to end of cycle 1 to.before start of cycle 2. End of C3 refers to start of cycle 2 to before start of cycle 4. End of cycle C6 refers to start of cycle 4 to before start of cycle 7. (**B**) Rate of best response (CR, CRi, MLFS) and corresponding MRD negative response (0.1% and 0.01% sensitivity), at remission and cumulative 3-month timepoint post-remission. C, cycle; CR, complete remission; CRi, complete remission with incomplete hematologic remission; MLFS, morphologic leukemia-free state.

**Figure 2 cancers-14-03576-f002:**
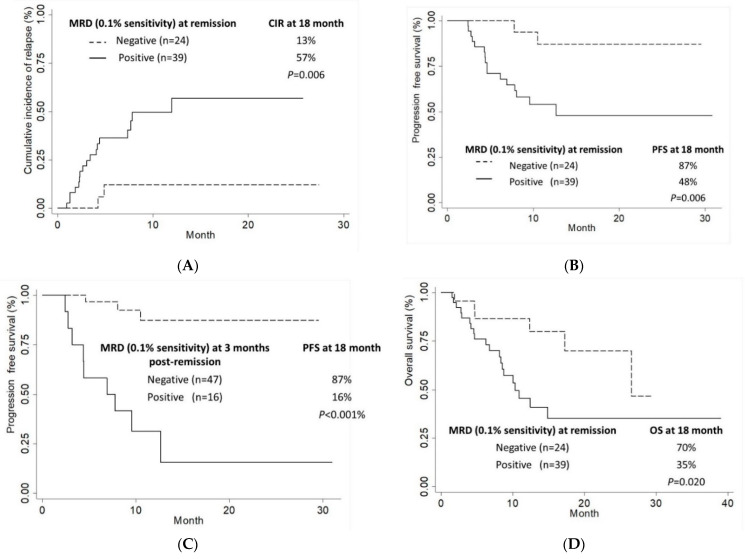
(**A**) Cumulative incidence of relapse (CIR), (**B**) Progression-free survival (PFS), (**D**) Overall survival (OS) based on MRD status at response (time of CR/CRi). (**C**) PFS based on MRD status at cumulative 3-month timepoint post-remission.

**Table 1 cancers-14-03576-t001:** Baseline characteristics of 63 responding patients with AML treated with azacitidine and venetoclax.

Characteristic	Patient Population (*n* = 63)	MRD ≥ 0.1% (*n* = 39)	MRD < 0.1% (*n* = 24)	*p*
Age ≥ 65, *n* (%)	36 (57%)	21 (58%)	15 (42%)	0.500
BM blasts, *n* (%)				0.487
<30%	26 (41%)	17 (65%)	9 (35%)	
30–50%	16 (25%)	8 (50%)	8 (50%)	
≥50%	19 (30%)	13 (68%)	6 (32%)	
Diagnosis, *n* (%)				0.433
De Novo	44 (70%)	25 (57%)	19 (43%)	
sAML with AHD	16 (25%)	12 (75%)	4 (25%)	
Therapy related	3 (5%)	2 (67%)	1 (33%)	
ELN 2017 risk group, *n* (%)				0.030
Favorable	11 (17%)	3 (27%)	8 (73%)	
Intermediate	20 (32%)	13 (65%)	7 (35%)	
Adverse	32 (51%)	23 (72%)	9 (28%)	
Complex cytogenetics, *n* (%)	17 (27%)	12 (71%)	5 (29%)	0.388
Mutations, *n* (%)				
NPM1	11 (17%)	4 (36%)	7 (64%)	0.055
FLT3 ITD/TKD	14 (22%)	7 (50%)	7 (50%)	0.416
Prior HMA	14 (22%)	12 (86%)	2 (14%)	0.038
Outcomes, *n* (%)				0.785
CR	15 (24%)	9 (60%)	6 (40%)	
CRi	42 (66.5%)	27 (64%)	15 (36%)	
MLFS	6 (9.5%)	3 (50%)	3 (50%)	
Mean time to best response, days (SD)	56 (34–71)	53 (30–64.5)	56 (40–106)	0.192
Transplant, *n* (%)	20 (32%)	11 (55%)	9 (45%)	0.441

Abbreviations: BM, bone marrow; sAML with AHD, secondary acute myeloid leukemia with antecedent haematological disorder; HMA, hypomethylating agents.

**Table 2 cancers-14-03576-t002:** Multivariate analysis of prognostic variables for the cumulative incidence of relapse (CIR), progression-free survival (PFS), and overall survival (OS).

Variables	CIR		PFS		OS	
	aSHR (95% CI)	*p* ^≠^	HR (95% CI)	*p* ^≠^	HR (95% CI)	*p* ^≠^
MRD ≥ 0.01% at response	4.70 (1.11–19.8)	0.035	4.62 (1.04–20.57)	0.044	1.99 (0.77–5.13)	0.156
MRD ≥ 0.1% at response	5.72 (1.33–24.64)	0.019	5.92 (1.34–26.09)	0.019	2.60 (1.02–6.63)	0.046
MRD ≥ 0.01% at cumulative 3-month post-remission	5.83 (1.13–29.93)	0.035	6.76 (1.31–34.73)	0.022	2.08 (0.65–6.64)	0.215
MRD ≥ 0.1% at cumulative 3-month post-remission	6.78 (2.77–41.50)	<0.001	14.55 (3.40–62.35)	<0.001	3.59 (1.21–10.71)	0.022

^≠^ Adjusted for ELN adverse versus favorable/intermediate risk group, complex karyotype versus normal karyotype and transplant; Abbreviations: aSHR, adjusted sub-hazard ratio.

## Data Availability

The data presented in this study, are available on request from the corresponding author. The data are not publicly available as they contain information that could compromise the privacy of research participants.

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
