# Peer review of "Real-World Experience of Measurable Residual Disease Response and Prognosis in Acute Myeloid Leukemia Treated with Venetoclax and Azacitidine"

_cancers, 2022, doi:10.3390/cancers14153576_

Round 1

Reviewer 1 Report

The authors report a retrospective analysis of 63 patients with newly diagnosed AML treated with azacitidine+venetoclax. They report significantly better outcomes in patients who achieved MRD levels <0.1%. Even though their assay could achieve sensitivities of 0.01%, these lower levels of MRD had less clinical important. Overall, this is a nice analysis of real world data which supports the existing literature on the impact of MRD with HMA+venetoclax-based combinations. I have the following specific comments:

1. The authors definition of RFS and OS are not standard. RFS should be from the time of remission (not start of therapy) until relapse or death from any cause. Similarly, OS should include death from any cause. What did the authors do if the patient died in remission. These must be counted as events in the RFS/OS analyses.

2. It is a bit hard to follow when patients achieved responses and then they occurred. I recommend a table showing best responses (CR, CRi, MLFS) and also the corresponding MRD (at 0.1% and at 0.01% cutoffs) at the 2 time points.

3. 90% of responders achieved CR/CRi. What did the other 10% achieve? MLFS? This should be explicitly stated.

4. The authors should provide MRD responses by cycle, or at least should provide the MRD response at end of cycle 1. Discussing MRD response in terms of months from start can be misleading since the duration of cycles can be variable between patients.

5. Section 3.2 suggests that all patients achieved "complete remission" but this is misleading, since a variety of responses are included.

6. Table 1 is presented in a bit of a confusing way. The header "MRD (0.01% sensitivity..." is out of place. The n (%) refer to the population of study. Also, in the MRD positive and MRD negative columns, the percentages should be relative to the characteristics. For example, the reader should be able to determine what was the MRD negativity rate among patients who achieved CR. As is, the table provides what percentage of MRD negative patients achieved CR, which is less clinically meaningful. 

7. As one of the major conclusions is that 0.1% is the best cutoff, the authors should show the outcomes for patients with MRD <0.01% vs 0.01% to 0.1% vs. >0.1%. This would help to make this point.

8. The authors need to clarify is the transplant OS analysis was performed as a landmark analysis.

9. Among the 5 patients who stopped treatment while MRD negative, after how long did they stop therapy. The authors provide median follow-up of these patients but it should specifically be median follow-up post-discontinuation. Similarly, the authors should clarify that the median 4.2 months in relapsed patients refers to post-discontinuation.

Reviewer 2 Report

In this manuscript, Ong and colleagues report their retrospective analyses of 63 patients with AML who received therapy with azacitidine and venetoclax off protocol at one of three centers in Singapore. They show that MRD by multiparameter flow cytometry at the time of response and after 3 months is associated with increased risk of relapse and shorter survival relative to patients without MRD at the same time points.

General comment: 

This is an interesting report that, while not providing any substantial new findings, confirms the data from the small series of studies that have assessed MRD in the setting of lower intensity therapy. Important limitations are the small cohort size (potentially one explanation why the cut-point of 0.01% did not yield statistically significant overall survival differences) and the use of 3 institutional MRD assays without any statement on similarities/differences between institutional assay characteristics. These limitations should perhaps more explicitly be highlighted in the discussion (and the methods section expanded with relevant information on inter-institutional differences in MRD testing – for example, have flow results been shared between institutions to determine whether there is consensus on MRD status and level of MRD?)

Additional comments:

1) Simple summary: the sentence “…we found that detectable residual disease by flow cytometry led to a higher incidence of relapse-free and shorter relapse-free and overall survival” is incorrect. Should be “higher incidence of relapse”. Would also suggest rephrasing and saying “was associated with” rather than “led” as causality cannot be inferred.

2) Introduction: “The clinically relevant MRD threshold likely depends on the intensity of the chemotherapy delivered…”. Should consider mentioning an alternative explanation, namely that the relevant MRD threshold depends on the characteristics of MRD test – especially when done via flow cytometry, techniques and expertise varies considerably, and it is very plausible that this impacts the threshold that proves prognostically most informative.

3) Multivariable models: stem cell transplant is included as a covariate. Please describe better whether this is history of stem cell transplant (i.e. factor present at baseline) or stem cell transplant performed subsequent to azacitidine/venetoclax treatment. In case of the latter, please describe in the methods section how exactly you accounted for this future risk factor as the typical multivariable cox regression model would be inappropriate to use.

4) In some parts of the results section, I was struggling understanding what exactly is meant with MRD-negative. Perhaps helpful if at all mentionings it is stated what the level of MRD is to call MRD negative.

Round 2

Reviewer 1 Report

The authors have adequately addressed my comments.